# Study on Microstructure and High Temperature Stability of WTaVTiZr_x_ Refractory High Entropy Alloy Prepared by Laser Cladding

**DOI:** 10.3390/e26010073

**Published:** 2024-01-15

**Authors:** Xiaoyu Ding, Weigui Wang, Haojie Zhang, Xueqin Tian, Laima Luo, Yucheng Wu, Jianhua Yao

**Affiliations:** 1Institute of Laser Advanced Manufacturing, Zhejiang University of Technology, Hangzhou 310014, China; dingxiaoyu0903@126.com (X.D.); 18970504374@163.com (W.W.); jxzhj@zjut.edu.cn (H.Z.); txq19802621829@163.com (X.T.); 2College of Mechanical Engineering, Zhejiang University of Technology, Hangzhou 310014, China; 3Collaborative Innovation Center of High-End Laser Manufacturing Equipment (National “2011 Plan”), Zhejiang University of Technology, Hangzhou 310014, China; 4School of Materials Science and Engineering, Hefei University of Technology, Hefei 230009, China; luolaima@126.com (L.L.); ycwu@hfut.edu.cn (Y.W.); 5Key Laboratory of Advanced Functional Materials and Devices of Anhui Province, Hefei 230009, China

**Keywords:** first wall material, high entropy alloys, laser cladding, microstructure, stability at high temperature

## Abstract

The extremely harsh environment of the high temperature plasma imposes strict requirements on the construction materials of the first wall in a fusion reactor. In this work, a refractory alloy system, WTaVTiZr_x_, with low activation and high entropy, was theoretically designed based on semi-empirical formula and produced using a laser cladding method. The effects of Zr proportions on the metallographic microstructure, phase composition, and alloy chemistry of a high-entropy alloy cladding layer were investigated using a metallographic microscope, XRD (X-ray diffraction), SEM (scanning electron microscope), and EDS (energy dispersive spectrometer), respectively. The high-entropy alloys have a single-phase BCC structure, and the cladding layers exhibit a typical dendritic microstructure feature. The evolution of microstructure and mechanical properties of the high-entropy alloys, with respect to annealing temperature, was studied to reveal the performance stability of the alloy at a high temperature. The microstructure of the annealed samples at 900 °C for 5–10 h did not show significant changes compared to the as-cast samples, and the microhardness increased to 988.52 HV, which was higher than that of the as-cast samples (725.08 HV). When annealed at 1100 °C for 5 h, the microstructure remained unchanged, and the microhardness increased. However, after annealing for 10 h, black substances appeared in the microstructure, and the microhardness decreased, but it was still higher than the matrix. When annealed at 1200 °C for 5–10 h, the microhardness did not increase significantly compared to the as-cast samples, and after annealing for 10 h, the microhardness was even lower than that of the as-cast samples. The phase of the high entropy alloy did not change significantly after high-temperature annealing, indicating good phase stability at high temperatures. After annealing for 10 h, the microhardness was lower than that of the as-cast samples. The phase of the high entropy alloy remained unchanged after high-temperature annealing, demonstrating good phase stability at high temperatures.

## 1. Introduction

Compared to traditional thermal power generation, controllable nuclear power generation is recognized as an advanced clean energy with high energy density and low carbon emissions, which potentially addresses the increasing energy demands with social development [1]. As the first wall is in contact with the reactants, the first wall material directly faces the impact of highly energetic heat flow and is severely eroded by high-throughput 14 MeV neutron irradiation and impurity elements such as hydrogen and helium [2,3]. In order to improve the performance of the first wall material in harsh environments, the construction material of the first wall should have good mechanical properties and radiation resistance at high temperatures [4,5]. Additionally, the elements that make up the material must be reduced activation, meaning that they are short-life or medium-life radioactive elements after several years of neutron irradiation with low radioactivity [6,7].

Nowadays, the first walls in most of the fusion reactors are constructed from zirconium Zr-based alloys. The performance of zirconium-based alloys is not strong enough to meet the material requirements. Extensive efforts have been made to develop advanced first wall materials for the next generation of reactors, including sodium-cooled fast reactors, lead-cooled fast reactors, supercritical water-cooled reactors, molten salt reactors, and ultra-high temperature gas-cooled reactors. The temperature and irradiation dose of these reactors have been improved, and the service conditions become even more severe. In a lead-cooled fast reactor, the first wall material needs to withstand 150 dpa irradiation and strong corrosion from liquid heavy metal Pb. Under such a service condition, Zr-based alloys would be seriously and quickly eroded. Irradiation-induced growth, high-temperature phase transition, high-temperature oxidation, and strong corrosion from liquid heavy metals [8,9,10,11] are major challenges Zr-based alloys are facing. Other first-wall materials, such as austenitic stainless steel, and Tiema Steel, are also vulnerable to plasma corrosion [12,13], irradiation swelling [14,15] and high-temperature stability [16,17,18]. Additionally, oxide dispersion strengthening (ODS) has also been employed to improve the creep resistance and radiation resistance of steel [19,20,21], but still not mature enough for mass production. Tungsten (W) has a high melting point, a high sputtering energy threshold, high thermal conductivity, low tritium retention, and low expansion rate; hence, it is one of the promising first wall material candidates. However, a ductile–brittle transition of W at high temperatures substantially impedes its application in fusion reactors [22]. Developing advanced nuclear cladding materials with excellent mechanical properties has emerged as a bottleneck to the development of nuclear reactors.

The multi-principle component design concept of high entropy alloys (HEAs) breaks through the traditional alloy design concept based on a single principal component [23]. High entropy alloys consist of five or more principal components with equimolar or near equimolar ratios, and the atomic fraction of each principal component is between 5% and 35% [24]. The feature of multi-principal elements in high-entropy alloys greatly enhances the mixing entropy of the alloy system, thereby reducing the Gibbs free energy, hinders the growth of intermetallic compounds, improves the mutual solubility of each element, and promotes the random distribution of atoms in the alloy in the lattice, eventually forming a robust, highly chaotic solid solution [25,26].

High entropy alloys have been rapidly developed in recent years due to their simple structure, excellent mechanical properties and convenient production. According to the nature and composition of the alloy, high entropy alloys can be divided into four categories: 3D-CoCrFeNi HEAs, lightweight HEAs, high entropy amorphous alloys, and refractory HEAs. Three-dimensional group HEAs are thoroughly studied and widely used because of their stable structure, simple manufacturing process and high performance–price ratio [27]. Lightweight high-entropy alloys have a low density and high specific strength, and are widely used in industries with strict quality control and performance requirements, such as transportation and aerospace [28]. High entropy amorphous alloys have a structure with disordered arrangement of metal atoms. A high entropy amorphous alloy usually has a large elemental distribution, good magnetic properties and amorphous forming ability [29]. Refractory high entropy alloys (RHEAs) are mainly composed of various refractory elements and have excellent high temperature mechanical properties, such as high strength, super hardness, good wear resistance, oxidation resistance, corrosion resistance, and high temperature resistance [30,31], etc. In particular, NbMoTaW RHEAs emerged recently as promising first wall materials due to their excellent phase stability and high temperature mechanical properties [32,33].

In this work, five elements of W, Ta, V, Ti and Zr were selected to design the composition of a high entropy alloy via a semi empirical formula based on the cocktail effect. High entropy alloys, WTaVTiZr, with varied compositions, improved purity, and high densities were deposited on the pure W substrate by a laser cladding method. The purity and density of the WTaVTiZr alloys were optimized via laser processing parameters. The influence of the Zr composition on the microstructure and phase composition of WTaVTiZr alloys was systematically studied and an optimal composition ratio of WTaVTiZr HEAs was obtained. The as-deposited WTaVTiZr HEAs were subjected to thermal annealing. The effects of thermal annealing on the microstructures, phase composition, and mechanical hardness of the WTaVTiZr HEAs were investigated and discussed.

## 2. Experimental Materials and Methods

Tungsten plates with a size of 100 mm × 100 mm × 10 mm (length × width × height) were used as the substrate for fabrication of WTaVTiZr HEAs. The chemical composition of tungsten plates is shown in Table 1. The cladding powders were prepared from five pure element powders of W, Ta, V, Ti and Zr (the purities are higher than 99.9%, and the average particle size is 48 μm). The microstructure of each element powder was characterized by SEM (scanning electron microscope). As shown in Figure 1, the five raw material powders revealed significant differences in particle size among the metal powders. The largest Ta powder exhibited a maximum particle size of 150 μm, while the smallest W powder particles measured approximately 1 μm. These substantial variations in particle size poses challenges for achieving homogeneous powder mixing. Furthermore, the low sphericity of Ti and Zr powders leads to decreased flowability, further complicating the uniform mixing of the metal powders.

The deposition of WTaVTiZr HEAs proceeded as following. A balance with an accuracy of 0.1 mg was used for powder measurement and composition design. Three elemental compositions were employed in mixed powders: WTaVTiZr equimolar alloy powder (P1), W_23_Ta_23_V_23_Ti_23_Zr_8_ (P2), and W_25_Ta_25_V_25_Ti_25_ (P3). The weighted powders with the designed composition were then poured into an SYH-15 three-dimensional motion mixer under argon atmosphere for mixing. An appropriate amount of WC balls were added for even mixing. The mixing parameters were a speed of 200~250 r/min and a mixing time of 5 h. The well-mixed powders with designed composition were placed in crucibles and dried at 80 °C for 3 h in a vacuum oven (Figure 2).

After 5 h of mixing, it was observed that V, Ti, Zr, and Ta, classified as strong carbide-forming elements, were more evenly mixed without any evidence of agglomeration. Conversely, the particle size-minimal element, W, displayed uniform distribution around the particles of the other four elements. This phenomenon indicates that the other elements could effectively encapsulate and disperse the W particles, resulting in a relatively homogeneous mixture.

The laser used in the experiment is a fiber coupled semiconductor laser (LDF6.000-40, Laserline, Germany) with a laser spot size of 4 mm. The laser cladding parameters for depositing HEAs with different Zr content are shown in Table 2. In order to eliminate the influence of surface stains and oxides on the cladding process, an angle grinder was used to polish the substrate surface until smooth without obvious stains. Anhydrous ethanol was then used for substrate cleaning, and the cleaned substrates were dried with a hairdryer. Then, a layer of the mixed powder was pre-coated on a W substrate, and laser cladding was carried out under argon atmosphere protection. The atmosphere protection device illustrated in Figure 3 comprises a sturdy aluminum plate at its bottom, ensuring both lightweight construction and resistance to deformation caused by heat. Acrylic plates are bonded together to form the device’s sides, facilitating internal observation and ensuring tightness. A heat-resistant, highly transparent plastic film connects the atmosphere protection box to the laser head, which is then sealed securely with heat-resistant adhesive tape. Small holes are provided for gas circulation within and outside the box. The lower opening is connected to an argon pipe to reduce oxygen content within the device, preventing oxidation reactions during fusion.

To analyze the surface morphology and microstructure of high-entropy alloy blocks with varying Zr contents, the following experimental approach and equipment were utilized.

The electric spark wire cutting machine was used to prepare three different overlay samples with varying Zr contents, with dimensions of 10 mm × 10 mm × 7 mm. These samples were labeled as Sample 1 (W_20_Ta_20_V_20_Ti_20_Zr_20_), Sample 2 (W_23_Ta_23_V_23_Ti_23_Zr_8_) and Sample 3 (W_25_Ta_25_V_25_Ti_25_). Each sample was polished using 80#, 240#, 400#, 800#, 1000#, 1200#, 1500# and finally 2000# sandpapers for rough and fine grinding. During the grinding process, water was used for cooling and lubrication to prevent debris from flaking off. After finishing with 2000# fine grinding, each sample underwent further polishing using a metallographic polishing machine, supplemented with the diamond polishing agent and water for cooling and lubrication. The resulting polished surface was cleaned with alcohol at regular intervals to remove dirt and dried with a blow dryer. The polished surface was then observed under an optical microscope for scratches, and any visible scratches were further polished until completely removed. Finally, the microstructure of the sample was analyzed under the microscope (OM, AXIO Scope. A1, Zeiss, Germany).

Using a field emission scanning electron microscope (FESEM, EVO18, ZEISS, Germany), the microstructure and morphology of the prepared sample can be observed and analyzed. Subsequently, the corresponding energy-dispersive spectroscopy (EDS) is employed to perform point and area scans, enabling the detection of elemental species and contents, as well as the distribution of elements in microscopic regions of the material.

The X-ray diffraction analyzer (XRD, D/MAX 2500 V) is employed for phase analysis of the sample prepared in this experiment. The experiment utilized Cu target Kα (λ = 0.154056 nm), with a voltage of 40 KV and a current of 40 mA. The scanning rate was set to 2°/min, the scanning angle range is between 10° and 100°. As there exists a one-to-one correspondence between each crystalline material and its diffraction pattern, this experiment combined XRD spectra with Jade 6 analysis software to determine the phase composition and crystal structure of the alloy sample.

To investigate the high temperature softening resistance of the WTaVTiZr high-entropy alloy, vacuum annealing was performed on original samples and annealed samples at 900 °C, 1100 °C, and 1200 °C for 5 h and 10 h using a muffle furnace and vacuum sealing techniques. The SEM, EDS, XRD, and microhardness results of the original samples and vacuum annealing samples at different temperatures and times were compared to study the high-temperature anti-softening performance and phase structure stability of this high-entropy alloy composition.

## 3. Results and Discussion

### 3.1. The Interface Morphology between WTaVTiZr High-Entropy Alloy and Matrix

Figure 4 depicts the cross-section of the cladding layer in a single-layer high-entropy alloy, where the cladding layer does not exhibit a metallurgical bond with the substrate. The cladding layer has a thickness of 0.6 mm and a width of 2.06 mm. Cracks have appeared on both sides of the cladding layer’s substrate, and pores have formed within the cladding layer. A preliminary analysis suggests that when the laser interacts with the surface of the substrate, it initially melts the high-entropy alloy powder pre-placed on the substrate’s surface. The constituent element powders of the high-entropy alloy have different melting points, and excessive energy input leads to the vaporization of the Ti element with a lower melting point, resulting in the formation of pores. The SEM results, depicted in Figure 5, reveal a distinct dendritic morphology in the single-layer cladding, with noticeable variations in dendrite sizes. However, after undergoing multiple layers of cladding, the microstructure gradually becomes more homogeneous, indicating a transition towards an equiaxed grain structure from the dendritic form.

EDS point scanning was performed on two types of specimens to analyze the elemental composition of the clad layer, as indicated in Table 3. The content of W element in the single-layer and multi-layer specimens was found to be 19.82% and 22.31%, respectively. The difference in element ratios from the previous composition design is minimal, suggesting that the substrate has a negligible impact on the clad layer of the high-entropy alloy.

### 3.2. Effect of Zr Content on Microstructure, Properties and Phase Structure of High Entropyalloy

The microstructure of high-entropy alloys with different Zr contents was further studied, as depicted in Figure 6. The analysis of the results reveals distinct characteristics for each sample. In Sample 1 (Figure 6a), the observed microstructure is primarily dendritic with a tendency towards equiaxed crystal formation. However, when compared with Sample 1, Sample 2 exhibits thicker dendrites and a more pronounced trend towards equiaxed crystal formation (Figure 6b). Nonetheless, Sample 2 also displays a significant presence of bubbles and holes. However, in Figure 6c, the microstructure of Sample 3 shows an initial stage of equiaxed crystal formation. As opposed to Samples 1 and 2, there is a gradual increase in the occurrence of bubbles and pores.

Figure 7 shows the elemental compositions at different positions of the specimens. In Figure 6a, the upper region of the dendrite for Sample 1 (locations c and d), there is a relatively high proportion of W and Ta elements, while Ti and Zr elements only constitute a small fraction. In contrast, Ti and Zr elements are enriched in the interdendritic region (locations a and b), which is the opposite of the upper dendritic region. The V element shows no significant variation in content either on the dendrite or in the interdendritic region. Overall, the elemental proportions for W, Ta, and V are 22.31%, 18.59%, and 17.24%, respectively, which deviate slightly from the designed composition. However, the Zr element content is as high as 28.03%, indicating a significant deviation from the original composition due to Zr macrosegregation in the alloy. The Ti element proportion is only 13.83%, which is also a large deviation from the design value and can be attributed to the excessive laser beam energy causing Ti element vaporization. In Figure 6b, the Ti element proportion is 9.70%, which deviates greatly from the original design value of 23%. The Zr content has decreased from the designed 8% to 4%, resulting from Zr macrosegregation and Ti element vaporization. As a result of the reduction in Zr and Ti elements, the relative proportions of Ta, V, and W elements increase. For Sample 3, the Ti element proportion is 9.69%, which deviates the most from the design value of 25%. SEM images indicate that the abundance of bubbles and voids is a result of Ti element vaporization. The higher laser power used in preparing Sample 3 compared to the other two samples has led to severe Ti element vaporization. Consequently, the remaining Ti element content in the sample is low, resulting in an increased proportion of the other elements.

According to the analysis of the phase structure of high-entropy alloys with different Zr contents (Figure 8), it can be observed that the diffraction peaks are the same for the three alloys, with only varying intensities. The strongest diffraction peaks for the three alloys are located at 2θ values of 39.618°, 57.249°, 71.808°, and 85.299°. The relationship between them, sin^2^θ_1_: sin^2^θ_2_: sin^2^θ_3_: sin^2^θ_4_, is 2:4:6:8, indicating compliance with the body-centered cubic (BCC) extinction rule. Therefore, the phase structure of high-entropy alloys with different Zr contents based on a pure W matrix is determined to be BCC. From the observation of the graph, it can be seen that as the Zr content increases, the intensity of the diffraction peaks decreases. This is attributed to the increased composition entropy as a result of the elevated Zr content, leading to severe lattice distortion and a weakening of the diffraction peak intensity. This finding is a result of considering the four effects of high-entropy alloys collectively [34].

Microhardness tests are conducted on high-entropy alloys with different Zr contents prepared, and the results were analyzed based on the data presented in Figure 9. The results show that the hardness values of the high entropy alloys prepared with W matrix are 616.4 HV for Sample 1, 611.4 HV for Sample 2 and 725.1 HV for Sample 3, which is a significant increase in the hardness of the alloys as compared to the hardness of the matrix (413.3 HV). However, there were notable differences in hardness between the high-entropy alloys with compositions of Zr (0%) and Zr (8%) compared to the alloy with a composition of Zr (20%). Furthermore, combined with the SEM testing results, it was observed that the high-entropy alloy with a composition of Zr (20%) exhibited a dendritic microstructure, whereas the microstructure of the alloys with compositions of Zr (0%) and Zr (8%) gradually transformed from a dendritic structure to an equiaxed structure. Additionally, numerous pores were observed within the microstructure. Therefore, it can be concluded that the microstructural characteristics of the alloy and the preparation process have significant effects on the micro-mechanical properties of high-entropy alloys.

### 3.3. Study of High Temperature Stability of WTaVTiZr High Entropy Alloy Annealed at Different Temperatures

In order to investigate the high-temperature performance of high-entropy alloys, the five-element equimolar ratio WTaVTiZr high-entropy alloy was subjected to annealing treatments at different temperatures and times. Firstly, the microstructure of the samples was analyzed. After annealing at 900 °C for 5 and 10 h, as shown in Figure 10a,b, it was observed that the microstructure appeared more uniform without any apparent defects such as cracks or pores. The microstructure was found to be similar to the initial microstructure depicted in the metallographic image before annealing. SEM results (Figure 11a,b) showed that the microstructure maintained its dendritic morphology even after annealing at 900 °C for 5 and 10 h, exhibiting no significant changes compared to the initial state. Additionally, EDS scans were conducted on the samples annealed at 900 °C for 5 and 10 h, and the results are summarized in Figure 12. According to the numerical data in the table, there were no notable variations in the element proportions and their distributions across different regions in the samples after annealing. Therefore, it can be inferred that the microstructure and element distribution of the samples remained largely unchanged after annealing at 900 °C for 5 and 10 h, indicating good high-temperature stability.

The annealing temperature was increased to 1100 °C, and the microstructure of the samples annealed at this temperature for 5 h and 10 h was tested. The samples annealed at 1100 °C for 5 h and 10 h were set as ‘Sample 4’ and ‘Sample 5’, respectively. Figure 10c shows the metallographic structure after annealing for 5 h, indicating a uniform structure without apparent pores. Extending the annealing time to 10 h, compared with the metallographic structure after 5 h of annealing, a black substance (Figure 10d) appeared in the interdendritic region of the metallographic structure. In order to further analyze the microstructure and element composition, SEM and EDS tests were conducted. The SEM results are shown in Figure 11c,d. It can be observed that the microstructure of Sample 4 is not significantly different from before annealing, while the microstructure of Sample 5 exhibits a relatively prominent black gully structure, corresponding to the black material in the metallographic structure. To understand the elemental composition of the black substance, EDS tests were performed on it, and the results are shown in Figure 13b. From the image, it can be seen that the brightness of Zr and Ti in the black region is significantly higher than in the gray region, while the brightness of W and Ta in the gray region is significantly higher than in the black region. It can be inferred that Zr and Ti elements are enriched between dendrites, while W and Ta elements are enriched on dendrites. EDS spot scans were performed on the two samples, and the results are summarized in Figure 12. The proportion of each element in Sample 4 was not significantly different from before annealing, and the degree of element segregation between dendrites and dendrites did not change significantly. The point scan results of the black part in Sample 5 show that the black part is mainly composed of Zr (66.75%), Ti (17.10%), and V (12.83%), with less content of W (0.91%) and Ta (2.41%). The proportion of elements in the gray part is different, with the Zr (66.75%) in the black part higher than the Zr (46.90%) in the gray part, while the Ti (35.22%) in the gray part is higher than the Ti (17.10%) in the black part. In summary, the microstructure and element composition of the sample did not change significantly after annealing at 1100 °C for 5 h, showing good high-temperature stability. When the annealing time of the sample was extended to 10 h, black substances appeared between dendrites, mainly enriched with Zr.

The experimental vacuum environment temperature was increased to 1200 °C. After different annealing times, the microstructure was examined and shown in Figure 10e,f, revealing a significant precipitation of black material. SEM and EDS tests were conducted on the two samples with different annealing times. The results of the SEM tests can be seen in Figure 11e,f, where a large amount of black substance can be observed between dendrites, mainly concentrated in the sparse area of dendrites. After annealing at 1200 °C, there is no significant change in the dendrite part. The element distribution of the samples before and after annealing for 10 h is studied, as shown in Figure 13a,c. The element distribution of the sample after 10 h of annealing is compared with that of the unannealed sample. It is found that the segregation of Zr and Ti elements in the interdendritic region became more pronounced after annealing (characterized by the brighter green regions), and the same trend is observed for the Ti element. EDS spot scanning tests are performed on the samples annealed at 1200 °C for different times, and the results are presented in Figure 12. It is observed that the Zr (approximately 50%) and Ti (approximately 27%) contents in the interdendritic region of the unannealed samples showed severe segregation after 10 h of annealing. The Zr content in the black part reached 62.20%, while the Ti content decreased to 22.69%. The most significant segregation is observed with a Zr content of up to 96.36% and a Ti content of 2.02%. The color difference between dendrites mainly arises from the variation in the content of Zr and Ti elements, and the black substance primarily consists of Zr-enriched material.

To investigate the phase stability of the samples after annealing at different temperatures for 5 h and 10 h, XRD tests are conducted on the annealed samples, and the results were consolidated. The findings are presented in Figure 14. It can be observed from the figure that no new diffraction peaks emerge after vacuum annealing, implying that no new phase is formed during annealing. The lattice constants of the high entropy alloys treated by different methods, from bottom to top, are as follows: 3.2168, 3.2124, 3.2049, 3.6759, 3.2180, 3.2036, 3.2043. By comparing these values, it is evident that the diffraction peak of the sample after annealing exhibits a slight shift towards the right. The lattice constant after annealing at 1200 °C for 10 h is reduced compared to the unannealed sample, from 3.2168 before annealing to 3.2043, indicating a relaxation in the structure of the phase. However, when considering the entire XRD test results, it can be concluded that the phase structure of the WTaVTiZr high entropy alloy with a five-element equimolar ratio remains stable after high-temperature annealing.

The micro-mechanical properties of the annealed samples under different conditions are analyzed and compared with the microhardness of the samples without annealing treatment, as shown in Figure 15. The microhardness of the samples annealed at 900 °C for 5 h and 10 h is 978.23 HV and 988.52 HV, respectively, with a small difference between them. When compared to the hardness of the unannealed sample at room temperature (725.08 HV), the high-entropy alloy exhibits remarkable stability at 900 °C and is strengthened in comparison to the alloy at room temperature. The microhardness of the samples annealed at 1100 °C for 5 h and 10 h is 930.47 HV and 864.65 HV, respectively. The hardness after 10 h of annealing is lower than that after 5 h, but it still surpasses the hardness of the unannealed sample. The microhardness of the samples annealed at 1200 °C for 5 h and 10 h is 798.75 HV and 686.35 HV correspondingly. The mechanical properties at this annealing temperature do not show significant improvements. Even after 10 h of annealing, the microhardness is lower than that of the unannealed sample. From the results of element segregation, it can be seen that with increasing temperature and prolonged annealing time, the solubility of Zr element in the solid solution phase decreases. Consequently, a large amount of black substance was enriched with Zr precipitates between dendrites, resulting in a weaker solid solution strengthening effect in the high-entropy alloy.

## 4. Conclusions

In this paper, a single-phase solid solution high-entropy alloy system with excellent radiation resistance is established based on five refractory and low-activation elements W, Ta, V, Ti and Zr, which provides technical reserves and theoretical basis for fusion reactor materials. The specific conclusions are as follows.

(I)The decrease in Zr content does not significantly affect the phase structure of the W_20_Ta_20_V_20_Ti_20_Zr_x_ high-entropy, which remains a single-phase body-centered cubic (BCC) structure. However, as the Zr content decreases, an increased number of pores can be observed within the alloy structure, accompanied by macrosegregation of elements. This suggests that reducing the Zr content leads to a more porous microstructure. Furthermore in microhardness. However, the hardness remains higher than that of the base alloy, indicating that the alloy with lower Zr content still possesses relatively high hardness specific to the W_20_Ta_20_V_20_Ti_20_Zr_x_ high-entropy alloy; the effects of changing Zr content may vary for different high-entropy alloy compositions or alloy systems.(II)The annealing study of the equimolar high-entropy alloy WTaVTiZr in a vacuum environment reveals that the annealing temperature impacts the material’s microstructure. As the temperature increases from 900 °C to 1200 °C, a gradually emerging black Zr element enrichment structure is observed, while no new phase structure is detected. This suggests that the phase structure of the alloy remains stable within the range of 900 °C to 1200 °C.(III)After undergoing high-temperature annealing treatment, the microhardness of the WTaVTiZr equiatomic ratio high entropy alloy can be enhanced, particularly at approximately 900 °C. However, annealing at higher temperatures may result in variations and reductions in microhardness.

## Figures and Tables

**Figure 1 entropy-26-00073-f001:**
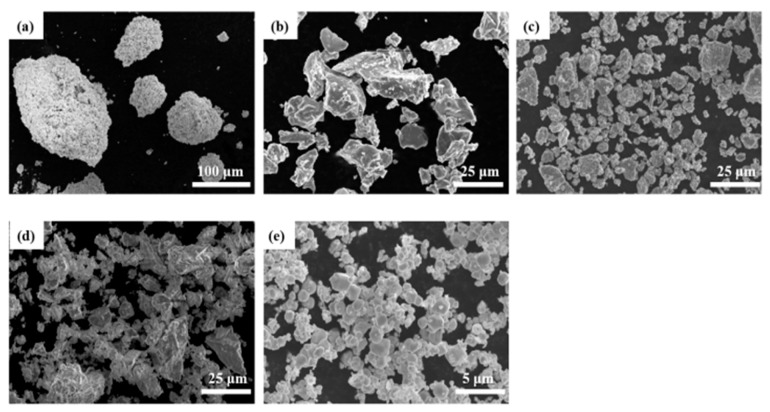
SEM images of raw material powder: (**a**) Ta; (**b**) Ti; (**c**) V; (**d**) Zr; (**e**) W.

**Figure 2 entropy-26-00073-f002:**
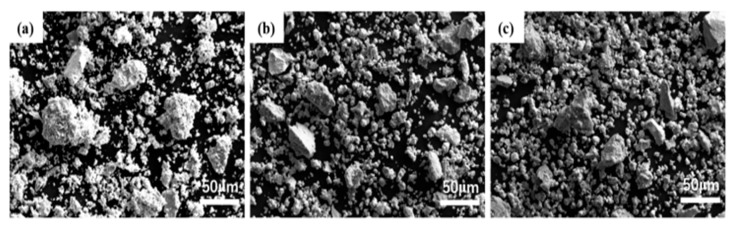
SEM images of high entropy alloy powder after powder mixing. (**a**) P1; (**b**) P2; (**c**) P3.

**Figure 3 entropy-26-00073-f003:**
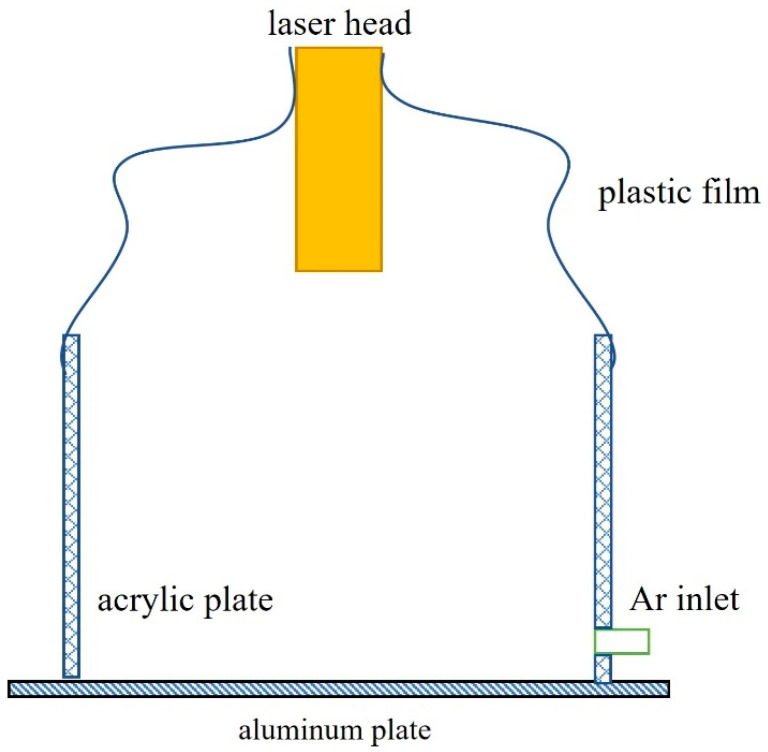
Atmosphere protection.

**Figure 4 entropy-26-00073-f004:**
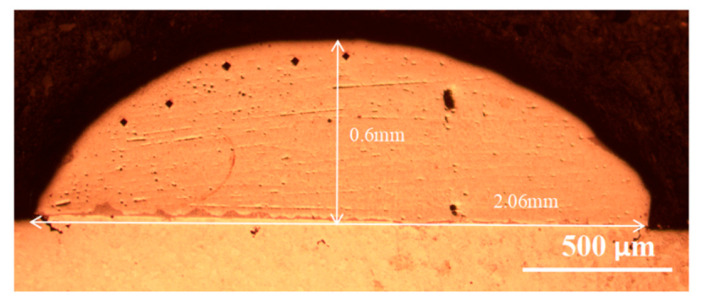
Cross section of single-layer high entropy alloy cladding prepared on W substrate.

**Figure 5 entropy-26-00073-f005:**
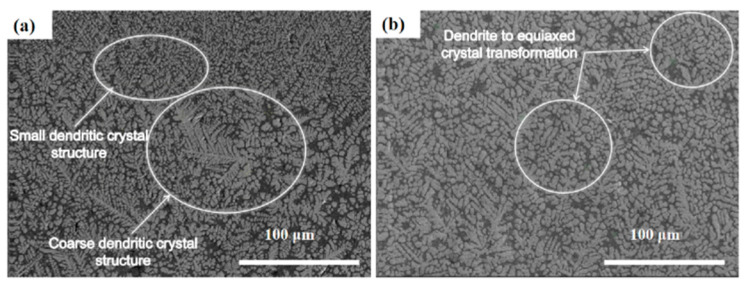
SEM image of high entropy alloy cladding layer prepared on W substrate. (**a**) Single pass cladding; (**b**) Multilayer cladding.

**Figure 6 entropy-26-00073-f006:**
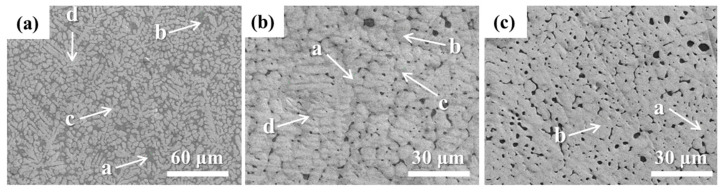
SEM image of high entropy alloy cladding layer prepared on tungsten substrate. (**a**) Sample 1; (**b**) Sample 2; (**c**) Sample 3.

**Figure 7 entropy-26-00073-f007:**
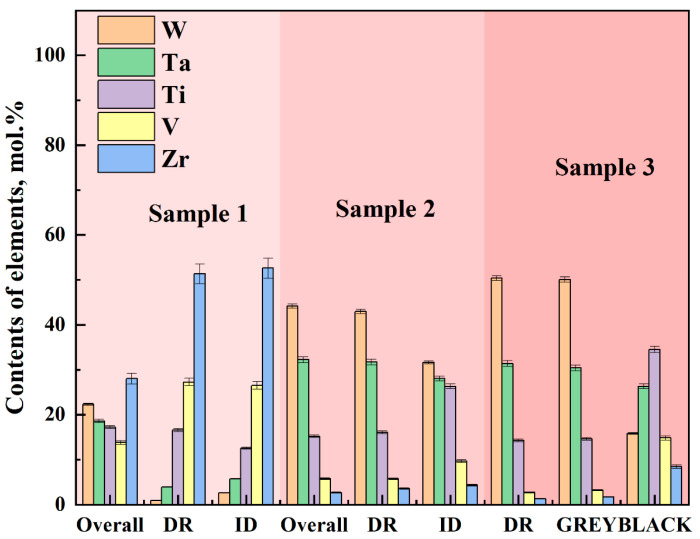
EDS of three component high entropy alloy cladding on tungsten substrate.

**Figure 8 entropy-26-00073-f008:**
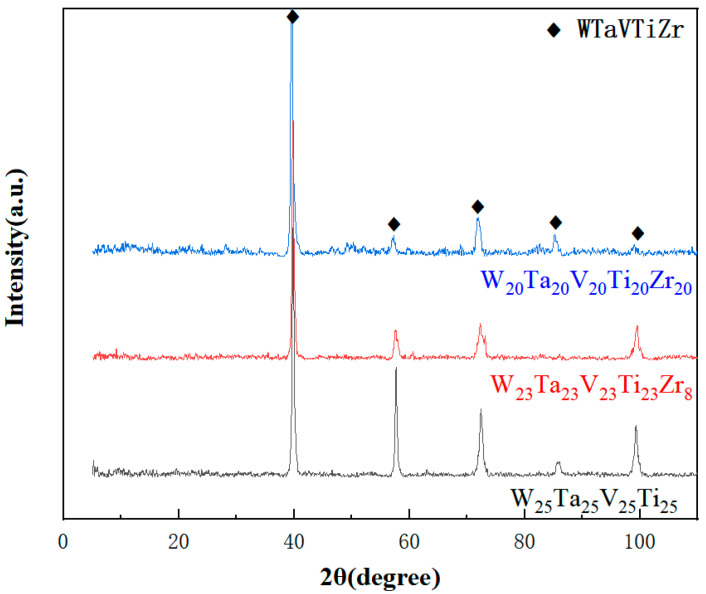
XRD images of three high entropy alloys on W substrates.

**Figure 9 entropy-26-00073-f009:**
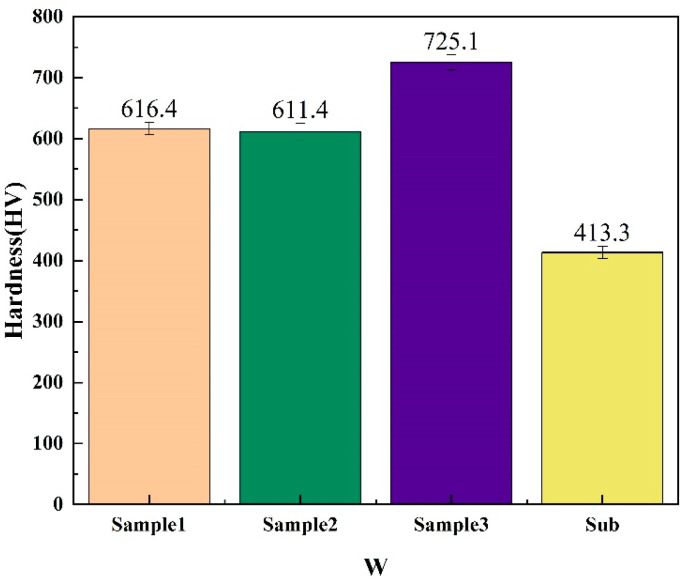
Microhardness of high entropy alloys.

**Figure 10 entropy-26-00073-f010:**
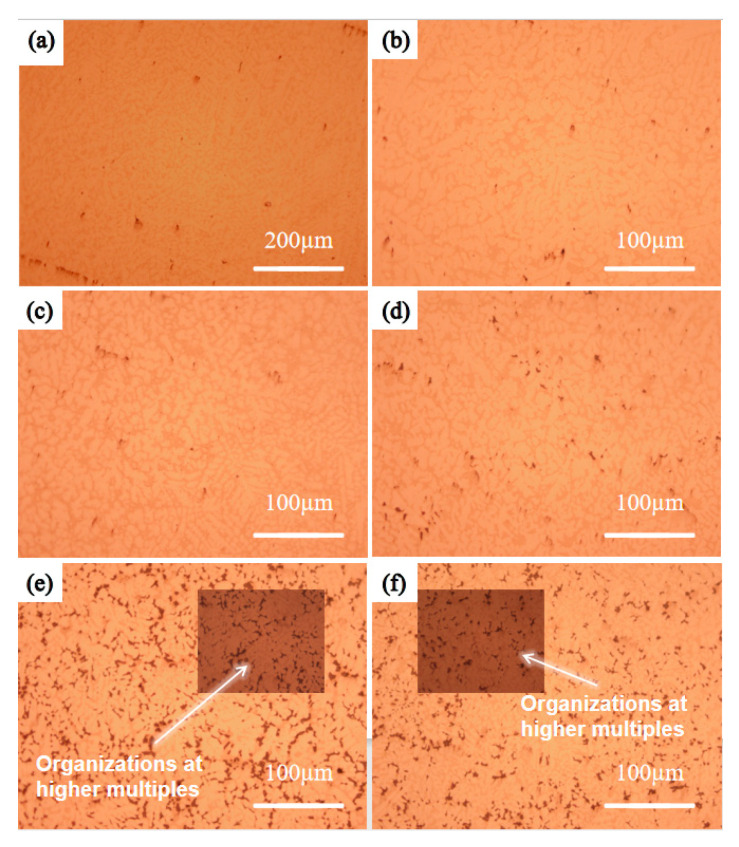
Metallographic diagram of vacuum annealed alloy in different temperature environments. (**a**) 900 °C, 5 h; (**b**) 900 °C, 10 h; (**c**) 1100 °C, 5 h; (**d**) 1100 °C, 10 h; (**e**) 1200 °C, 5 h; (**f**) 1200 °C, 10 h.

**Figure 11 entropy-26-00073-f011:**
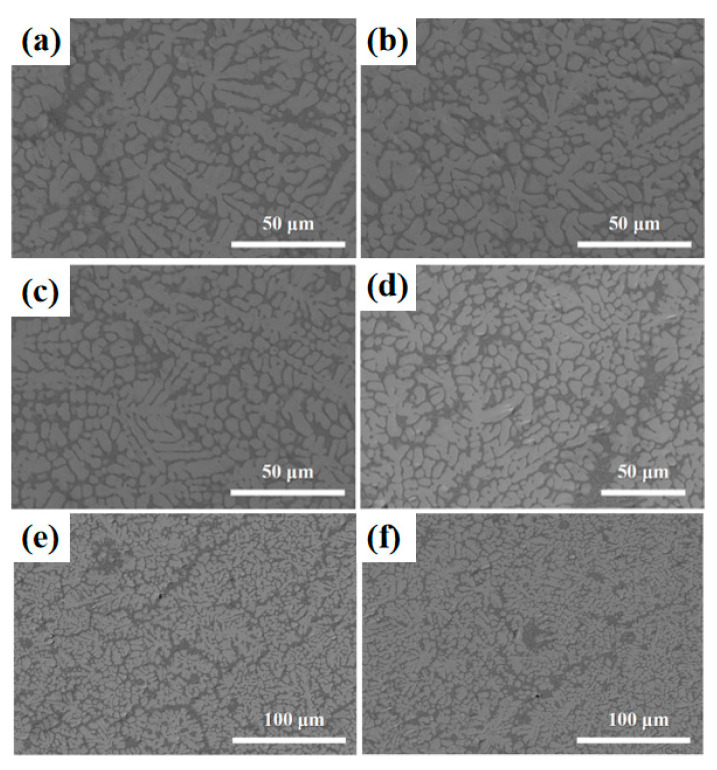
SEM images of high entropy alloy after 900 °C; 1100 °C and 1200 °C vacuum annealing. (**a**) 900 °C, 5 h; (**b**) 900 °C, 10 h; (**c**) 1100 °C, 5 h; (**d**) 1100 °C, 10 h; (**e**) 1200 °C, 5 h; (**f**) 1200 °C, 10 h.

**Figure 12 entropy-26-00073-f012:**
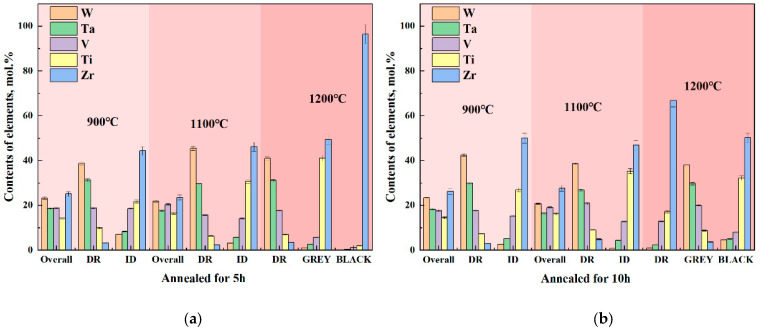
EDS test results of high entropy alloy after vacuum annealing at different temperatures, (**a**) annealed for 5 h, (**b**) annealed for 10 h.

**Figure 13 entropy-26-00073-f013:**
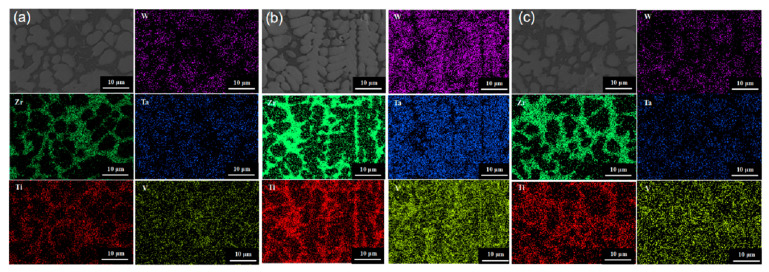
EDS surface scan images of alloy in different states. (**a**) No vacuum annealed; (**b**) Vacuum annealed at 1100 °C for 10 h; (**c**) Vacuum annealed at 1200 °C for 10 h.

**Figure 14 entropy-26-00073-f014:**
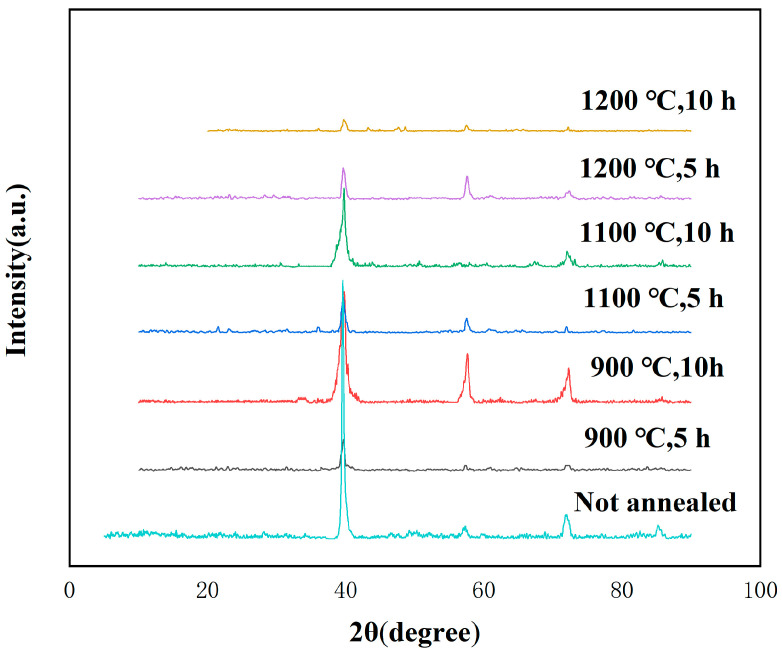
XRD image of high entropy alloy after vacuum annealing.

**Figure 15 entropy-26-00073-f015:**
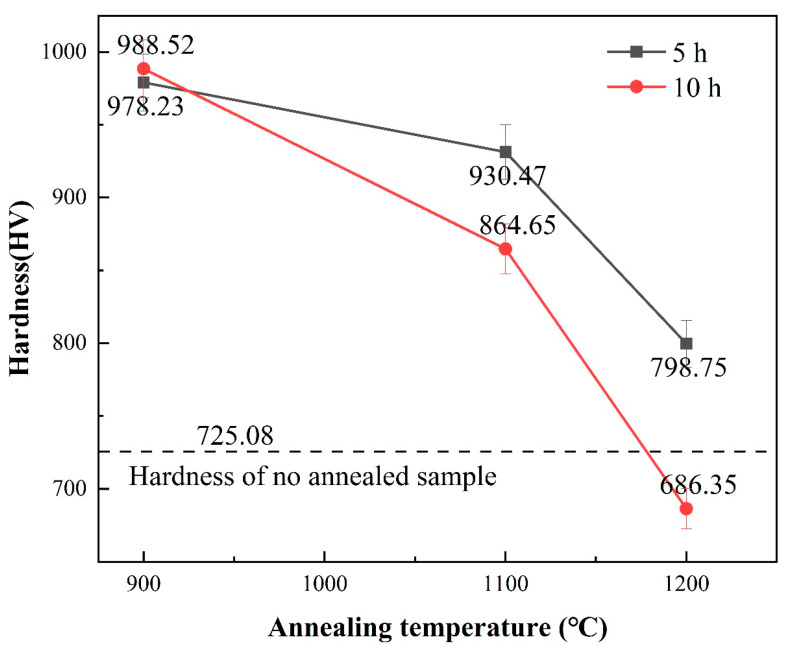
Microhardness diagram of high entropy alloy after vacuum annealing.

**Table 1 entropy-26-00073-t001:** Chemical composition of tungsten matrix.

Elements (wt.%)								
C	Ca	Ni	Mg	Fe	Mo	Sb	Si	W
0.06	0.04	0.06	0.04	0.05	0.1	0.1	0.1	99.45

**Table 2 entropy-26-00073-t002:** Laser cladding parameters of multi-layer WTaVTiZr HEAs on W substrates.

	Power, W	Spot Diameter, mm	Powder Thickness, mm	Scanning Speed, mm/s
W_20_Ta_20_V_20_Ti_20_Zr_20_	3300	4	0.7	5
W_23_Ta_23_V_23_Ti_23_Zr_8_	3300	4	0.7	5
W_25_Ta_25_V_25_Ti_25_	4300	4	0.7	5

**Table 3 entropy-26-00073-t003:** EDS test of high entropy alloy cladding layer prepared on W substrate.

	W (%)	Ta (%)	V (%)	Ti (%)	Zr (%)
Design	20	20	20	20	20
Single pass cladding	19.82	15.18	18.24	19.40	27.37
Multilayer cladding	22.31	16.32	19.51	13.83	28.03

## Data Availability

Data are contained within the article.

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
