# Peer review of "Study on Microstructure and High Temperature Stability of WTaVTiZrx Refractory High Entropy Alloy Prepared by Laser Cladding"

_entropy, 2024, doi:10.3390/e26010073_

Round 1
Reviewer 1 Report
Comments and Suggestions for Authors
The manuscript investigates the influence of Zr element content on the microstructure, phase composition, hardness, and phase stability of WTaVTiZr refractory HEA prepared through laser cladding technique. However, I think the manuscript is not well designed. The results cannot sufficiently support the conclusions. Detail comments are shown as follows,
1. The scale has not been included in Figure 6.
2. In the introduction section. Line 18, “WTaVTiZrx, with low activation and high entropy was theoretically designed based on …” What does the low activation refer to? Similarly, the 3D-HEAs may be refer to the CoCrFeNi series HEAs? The complete abbreviation should be provided.
3. The scanning rate for laser cladding should be provided in Table 2.
4. As shown in Fig.5, is there any criteria to judge the column grain and equiaxed grains?
5. I think there are some other phases in the as-cast WTaVTiZr alloy as indicated by the XRD patterns.
6. Section 3.1, “Effect of annealing at different temperatures on high temperature mechanical properties of WTaVTiZr high entropy alloy”. I have not seen the high temperature mechanical properties results in the manuscript, such as high temperature tensile or creep tests.
7. As the annealing temperature increases, the hardness increases firstly and then decreases, the change in hardness may not be fully explained by the decrease in solubility of the solids alone. The investigated WTaVTiZr refractory HEA should have a high melting point. The 900 ℃ annealing is too low for phase separation. The definition of entropy is “G=H-TS”, at the higher temperature the entropy plays the dominate role on the formation of solid solution, while at the intermediate temperature the phase separation becomes more apparently. At 900 ℃, the temperature is too low for phase separation due to the limitation of dynamic factors. Therefore, the detail microstructural analysis should be provided.
8. Fig. 12. The intensity of XRD peaks cannot show the distortion of lattice parameter. There are too many factors to affect the intensity.
Comments on the Quality of English LanguageThe MS should be checked carefully.
Reviewer 2 Report
Comments and Suggestions for Authors
This manuscript studied the microstructure and high temperature stability of WTaVTiZrx refractory high entropy alloy rrepared by laser cladding. The article is generally innovative, and there are many unrigorous expressions. Therefore, an major revision is recommended.
1. In lines 28 to 29 of the abstract, the author states, ", the microhardness of alloy samples annealed at 900°C or 1100°C for 5 reaches up to 988.52 HV." The unit for "5" needs to be provided, and 988.52 HV is a specific value, but the context is "at 900°C or 1100°C," is this reasonable?
2. “The effects of thermal annealing on the microstructures, phase
composition and mechanical hardness of the WTaVTiZr HEAs were investigated and discusse” needs to be corrected to “The effects of thermal annealing on the microstructures, phase composition, and mechanical hardness of the WTaVTiZr HEAs were investigated and discussed.”
3. “The microstructure of each element powder was characterized by SEM (Fig.1). Analysis of scanning electron microscope (SEM) images of the five raw material powders revealed significant differences in particle size among the metal powders.” The full name of a proper noun such as SEM should be given the first time it appears in the full text.
4. The author mentioned earlier, "The cladding powders were prepared from five pure element powders of W, Ta, V, Ti, and Zr (the purities are higher than 99.9%, and the particle sizes are about 48 μm)." However, they later described, "The largest Ta powder exhibited a maximum particle size of 150 μm, while the smallest W powder particles measured approximately 1 μm." The information about particle sizes seems to be confusing.
5. The caption for Figure 1 contains a repetition.
6. “Conversely, the particle size-minimal element, W, displayed uniform distribution around the particles of the other four elements.”
How to determine that the other four elements are uniformly distributed when there is no way to distinguish the four elements in Figure 2.
7. The hardness in Figure 8 needs to be supplemented with some values to draw error bars when drawing the diagram. The same problem is present in Figure 13.
8. The reason for the disappearance of bubbles and pores after annealing of the sample needs to be specified.
9. The XRD analysis showed that the annealing did not change the phase structure of the alloy, but a black Zr element enrichment structure appeared. Since XRD may not be able to clearly show the phase with a small content, it is recommended to confirm the phase structure of the black matter by TEM.
10. The authors emphasize that the microstructure of the alloy is improved by annealing, but the hardness is reduced, the reason for which is not given.
Reviewer 3 Report
Comments and Suggestions for Authors
For their work, the authors design the composition of a high entropy alloy by semi empirical formula based on cocktail effect. They were studying the effect of Zr additions on the high entropy alloy and then the alloy was deposited on pure W substrate by a laser cladding method. They explain in good detail many aspects of experimental development. On the other hand, there are a very good presentation, discussion and analyses of the obtained results. Which made it possible to write sound conclusions. I recommend the publication of the article, after considering the following suggestions to it:
To authors
1. Why was Zr chosen as a study element?
2. The authors indicated that the mixing parameters were: speed of 20 ~ 25 r/min and the mixing time of 5 h. I consider that these mixing times are too short and do not guarantee a good homogenization of the elements. Evidence of this is the cracking that the authors indicate is present in the coating layer. Can you please explain why you chose these mixing times?
3. In figure 2 there are observed a large dispersion of particle sizes, how can homogeneous coatings and even more so the formation of HEA be ensured with these sizes? Would you explain these please?
4. In section 3.1 and based on the observation in figure 4, the authors say that the cladding layer does not exhibit a metallurgical bond with the substrate. Whether this is good or not?, if there is not a good bond between the substrate and the coating, I would think that the coating was not good.
Reviewer 4 Report
Comments and Suggestions for Authors
The paper studies the Zr role in WTaVTiZr high entropy alloys. The material is suitable for application in the reactor’s first wall in a nuclear power generator. The interest in this topic is always high, and it deserves to be studied. The Authors present three different HEA Zr stoichiometries synthesized by laser cladding on a tungsten foil substrate, analyzing the morphological, chemical, structural, and mechanical properties. The results presentation and discussion are often confusing, and they do not clearly support some conclusions. The paper can not be published in this form. Several revisions are necessary, more in detail:
1. Introduction: The number of references must be improved, considering the topic's importance.
2. Experimental) Table 1 shows the substrate composition and not the HEAs one. Does the cited particle size of 48um refer to Zr only, or is it a medium value for all elements, as the real size is very different from 48um? Concerning the results after mixing, how did the Authors identify the elements in the powders? After the grinding polishing, what was the surface roughness? The Authors must improve the description of Fig.3 (what is the red layer? The inlet is for Ar and not air), in the main text and in the caption. The error for the EDS analysis must be shown.
3. Results, 3.1: the claim about multiple cladding layers properties (page 12) is not clear, being referred to in Fig.5 where a single layer HEA is shown; a SEM picture from a multilayer cladding is needed.
4. Results, 3.2: Comparing the Ti stoichiometry in samples 1 and 2, having the same laser power, the value is lower for the sample having the higher nominal Ti percentage; the Authors must comment on this issue, as on the intensity of the main XRD peak, which doesn’t reduce with Zr content. The Authors should comment on the Ta and Ti percentages differences (table 3). Microhardness: The authors must specify the different sample values; the use of “respectively” is not enough (page 17).
5. Results, 3.3: The Authors never cite which samples they annealed, nor in the main text, nor in the figures. Probably the Zr25% one, as can be deduced from table 5, or the equimolar one (see Conclusions). Anyway, they annealed only one HEA type, and this must be clearly claimed. In general, the description suffers poor clarity, e.g., samples 1 and 2 are what? A figure showing the different stoichiometries vs the annealing temperature is necessary instead of the unintelligible Table 5 (where one Zr value is missing). XRD: Fig.12 shows several peak lineshape (all cases) and position (1100°C, 5h) changes, suggesting not negligible structural modifications that the Authors did not discuss, leading to subsequent phase changes.
6. Conclusions: The Authors must briefly introduce their work before going into details of conclusions. Considering the final application requirements of these materials, the Authors must comment on their use in reactor’s first wall, considering the real working conditions. The Authors should also compare their results with other studies, including theoretical works.
7. Table 2: readability must be improved.
8. Figure 6: the length scale is not shown. The Authors must describe the different pointers, also in the main text (page 14), where no references to Fig.6 are made. The sample 1-3 labeling must be inserted in the caption. An EDS mapping, the same as in Figure 11, is required to improve the discussion.
9. Figure 13: the presence of the three symbols of the not annealed sample at different temperatures is misleading. A single horizontal line must be used.
10. Table 4: The authors must include the different sample 1-3 labeling also in the table to improve clarity. The Authors should switch the V and Ti column positions according to the material formula and previous results in Table 3.
11. Some typos and errors are present. The punctuation must be improved.
Comments on the Quality of English LanguageEnglish quality is almost acceptable, but several typos must be corrected.
Round 2
Reviewer 1 Report
Comments and Suggestions for Authors In my opinion, the article does not meet the publication criteria due to the apparent shortcomings. For instance, during higher temperature annealing of refractory high entropy alloys, phase segregation may occur. Based on the SEM image provided (Fig.14), it is difficult to ascertain the solid solution structure. Actually, after 1200 ℃/10 h annealing, clear phase separation can be observed according to the XRD results. However, this phenomenon seems to have been overlooked by the authors.Author Response
Please see the attachment.

Reviewer 2 Report
Comments and Suggestions for Authors
After revisions, the article is recommended for publication.
Author Response
Thank you for graciously consenting to the publication of this article.
Reviewer 3 Report
Comments and Suggestions for Authors
They manuscrito was corrected very well, authors have attended allá sugestiones. Paper can ve published
Author Response

(The authors gave the same response as above.)

Reviewer 4 Report
Comments and Suggestions for Authors
The Authors partially revised the paper according to my previous comments. However, they did not address XRD analysis issues (points 4 and 5). In particular, the Authors must smooth the claim “the WTaVTiZr high entropy alloy with a five-element equimolar ratio remains stable after high-temperature annealing”, inserting “almost stable”, as the results in Fig.14 suggest some not negligible changes, partially described on page 23. Also, the Conclusions must be changed accordingly (point II).
Comments on the Quality of English LanguageEnglish quality is acceptable
Round 3
Reviewer 1 Report
Comments and Suggestions for Authors
It can be accepted in this version